# The First Whole Genome Sequence and Methylation Profile of *Gerronema lapidescens* QL01

**DOI:** 10.3390/jof11090647

**Published:** 2025-09-01

**Authors:** Yanming Qiao, Zhiyuan Jia, Yuying Liu, Na Zhang, Chun Luo, Lina Meng, Yajie Cheng, Minglei Li, Xiuchao Xie, Jianzhao Qi

**Affiliations:** 1Shaanxi Province Key Laboratory of Bio-Resources, Qinba State Key Laboratory of Biological Resources and Ecological Environment (Incubation), School of Biological Science and Engineering, Shaanxi University of Technology, Hanzhong 723001, China; yanmingq1990@163.com (Y.Q.); 15129091626@163.com (C.L.); menglina556@163.com (L.M.); 15194404742@163.com (Y.C.); 2Shaanxi Key Laboratory of Natural Products & Chemical Biology, College of Chemistry & Pharmacy, Northwest A&F University, Yangling 712100, China; 2023014689@nwafu.edu.cn (Z.J.); lyy0905@nwafu.edu.cn (Y.L.); zhangnn2402@163.com (N.Z.); 3Center of Edible Fungi, Northwest A&F University, Yangling 712100, China; mlli@nwafu.edu.cn

**Keywords:** *Gerronema lapidescens*, methylome, medicinal fungi, sclerotium-forming fungi

## Abstract

*Gerronema lapidescens* (Lei Wan), a valued medicinal basidiomycete traditionally employed for antiparasitic and digestive ailments, faces severe conservation threats due to unsustainable wild harvesting and the absence of reliable cultivation protocols. To address this crisis and unlock its pharmacotherapeutic potential, we present the first chromosome-scale genome assembly and comprehensive methylome profile for the wild strain *G. lapidescens* QL01, domesticated from the Qinling Mountains. A multi-platform sequencing strategy (Illumina and PacBio HiFi) yielded a high-quality 82.23 Mb assembly anchored to 11 chromosomes, exhibiting high completeness (98.4% BUSCO) and 46.03% GC content. Annotation predicted 15,847 protein-coding genes, with 81.12% functionally assigned. Genome-wide analysis identified 8.46 million high-confidence single-nucleotide polymorphisms (SNPs). Notably, methylation profiling revealed 3.25 million methylation events, with elevated densities on chromosomes 4, 9, and 10, suggesting roles in gene silencing and environmental adaptation. Phylogenomic analyses clarified the evolutionary status of *G. lapidescens*, whilst gene family evolution indicated moderate dynamics reflecting niche adaptation. Carbohydrate-Active enzymes (CAZymes) analysis identified 521 enzymes, including 211 Glycoside Hydrolases (GHs), consistent with organic matter degradation. Additionally, 3279 SSRs were catalogued as molecular markers. This foundational resource elucidates *G. lapidescens*’s genetic architecture, epigenetic regulation, evolutionary history, and enzymatic toolkit, underpinning future research into medicinal compound biosynthesis, environmental adaptation, germplasm conservation, and sustainable cultivation.

## 1. Introduction

The fungus *Gerronema lapidescens* (Basidiomycota, Marasmiaceae) produces small fruiting bodies in temperate and subtropical broadleaf and coniferous forests [1]. It forms mycorrhizal associations with certain trees and typically develops sclerotia in decaying wood or soil [2]. Taxonomic classification has been contentious: historically placed in *Omphalia* (Agaricaceae) or Polyporaceae, it was reclassified into *Gerronema* by Singer for omphalinoid, lignicolous fungi [3] with small-to-medium clitocyboid fruiting bodies. Integrated morphological and molecular phylogenetic analyses by Zhang et al. subsequently established the current nomenclature, *Gerronema lapidescens* (Horan.) Ming Zhang & W.X. Zhang [1].

Documented as ‘Lei Wan’ in the Shennong Bencao Jing [4] and classical materia medica (e.g., Compendium of Materia Medica, Zhonghua Bencao), *G. lapidescens* is characterised as cold-natured and sweet-tasted. Traditionally used for antiparasitic and digestive purposes—particularly against parasitic abdominal pain and infantile malnutrition with food retention [5,6]—its millennia-long application is supported by modern pharmacology. Bioactive compounds include *G. lapidescens* lectin and sterols [7,8], conferring antitumour, antiparasitic, anthelmintic, anti-inflammatory, antiviral, immunomodulatory, and hypoglycaemic activities [9]. Rising demand for this promising anticancer source contrasts sharply with supply constraints: reliance on wild resources, protracted growth cycles, and stringent environmental requirements have precipitated population decline, market shortages, and price inflation [10]. Despite advances in cultivating other wild fungi, reliable *G. lapidescens* cultivation remains elusive, perpetuating wild harvesting and natural resource depletion.

Genome sequencing offers distinct advantages over traditional fungal identification: it enables precise molecular delineation, phylogenetic resolution, and exploration of evolutionary history. Critically, it reveals metabolic networks, biosynthetic pathways, and nutritional requirements across growth stages. Methylation modifications in promoter regions can silence core genes by hindering transcriptional initiation, reducing synthesis of key metabolic enzymes and impairing growth. As a pivotal epigenetic mechanism, methylation facilitates rapid environmental adaptation without altering DNA sequences. Recognising the instrumental role of genomics in advancing cultivation, this study undertook whole-genome and methylome analyses of *G. lapidescens* QL01 from the Qinling Mountains. The purpose is to investigate its genetic architecture, metabolite composition, and adaptation mechanisms by (1) elucidating genomic traits and metabolic networks across growth stages; (2) identifying genes governing medicinal compound biosynthesis and environmental adaptation; (3) pioneering in silico prediction of core gene methylation—a key epigenetic regulator that may silence metabolic enzymes via promoter modifications. These foundations will catalyse new research avenues to expand the practical use of *G. lapidescens*, particularly in overcoming cultivation barriers and sustainable resource utilisation.

## 2. Materials and Methods

### 2.1. Fungal Material and Nucleic Acid Extraction

The fresh sclerotial tissues of wild *G. lapidescens* were collected for tissue isolation, and the mycelium was obtained on PDA (Potato Dextrose Agar) medium. Based on the morphological characteristics of the mycelium and the ITS sequence features, it was identified as *G. lapidescens*. The high-quality mycelia of *G. lapidescens* QL01 were cultured in potato dextrose broth (PDB) liquid medium (25 °C, 180 rpm, one week) to obtain mycelium for DNA extraction. Genomic DNA was isolated using a Fungal Genomic DNA Rapid Extraction Kit (Sangon Biotech, Shanghai, China). Integrity was assessed via agarose gel electrophoresis; purity (OD_260_/_280_ = 1.8–2.0) was verified using a Nanodrop spectrophotometer (Thermo Fisher, Waltham, MA, USA). For Illumina sequencing, DNA (≥5 µg, ≥20 ng/µL) was fragmented (~300 bp) using a Covaris ultrasonicator (Covaris, Woburn, MA, USA). Libraries were constructed with the NEBNext^®^ Ultra™ DNA Library Prep Kit (NEB, Ipswich, MA, USA), followed by sequencing on an Illumina NovaSeq platform. For Hi-C sequencing, chromatin was cross-linked with formaldehyde and digested with HindIII. Hi-C libraries (300–700 bp inserts) were quantified by Qubit™ 2.0 Fluorometer (Invitrogen, Waltham, MA, USA), and qPCR-validated before PE150 sequencing on an Illumina NovaSeq 6000. For Oxford Nanopore sequencing, DNA (≥10 µg, ≥80 ng/µL) was fragmented (Megaruptor^®^, Diagenode, Denville, NJ, USA), size-selected (BluePippin^®^, Sage Science, Beverly, MA, USA), and processed using the SQK-LSK109 kit (Oxford Nanopore Technologies, Oxford, UK).

### 2.2. Genome Assembly and Chromosome Assembly

Genome size and complexity were estimated via k-mer (21-mer) analysis of Illumina reads using GenomeScope 2.0 [11]. All parameters were kept at their default settings throughout the analysis. Initial assembly employed NECAT [12], followed by two rounds of error correction with Racon v1.4.1 (ONT data) and Pilon v1.23 (Illumina data) [13]. Assembly completeness was assessed using CEGMA 2.5 [14] and BUSCO v4.0 [15]. For chromosome-level assembly, adapters and low-quality reads were trimmed [16]. HiFi reads were assembled (Hifiasm), contigs aligned (BWA v0.7.10-r789) [17], and scaffolds evaluated for continuity, accuracy, consistency, and completeness [16,17].

### 2.3. Genome Annotation

For protein-coding gene annotation, this study employed an integrated approach combining de novo prediction, homology-based searches, and transcriptome-guided methods. For de novo prediction, gene models were predicted using Augustus (v3.1.0) [18]. In the construction of reference gene models, GeMoMa (v1.7) was utilised as a tool for homology-based annotation, drawing upon genomic data from multiple *Agaricales* species, including *Lentinula edodes* and *Dendrothele bispora.* The workflow of the annotation process based on the transcriptome is outlined as follows. The RNA-seq data were then mapped to the reference genome using hisat (v2.1.0), followed by transcript assembly with StringTie (v2.1.4). Subsequently, gene prediction was performed on the assembled transcripts using GeneMarkS-T (v5.1). Furthermore, PASA (v2.4.1) was utilised to predict genes from a library comprising full-length PacBio transcripts and Trinity (v2.11)-assembled unigenes. Gene models derived from these approaches were integrated using EVM (v1.1.1) to generate the final coding gene annotations. The subsequent functional annotation of protein-coding genes was conducted through the utilisation of BLASTP searches against a multitude of public databases, encompassing EggNOG, GO, KOG, Pfam, TrEMBL, NCBI NR, SwissProt, and KEGG v20191220. The non-redundant protein sequences from NCBI, Swiss-Prot, COG, and KEGG were employed as queries in this process. All alignments were performed with an E-value cutoff of 1.0 × 10^−5^.

The present study employed a suite of bioinformatic tools for the systematic identification of non-coding genes, encompassing miRNA, rRNA, tRNA, snoRNA, and snRNA. The prediction of tRNA genes was performed using the tRNAscan-SE programme (v1.3.1), which is based on the analysis of eukaryotic genomic characteristics. The annotation of ribosomal RNA genes was achieved by utilising the Barrnap (v0.9) tool. The identification of miRNAs was conducted in accordance with the guidelines set out in the miRBase 21 database. Prediction of snoRNAs and snRNAs integrated resources from the Rfam database (v14.5) was performed employing INFERNAL (v1.1) [19]. The identification of pseudogenes was achieved through the implementation of a combined algorithm, incorporating GenBlastA (v1.0.4) and GeneWise (v2.4.1).

The study selected a comparative dataset comprising 16 representative basidiomycete species, including *G. lapidescens* QL01, for the systematic analysis of CAZymes. The functional annotation and taxonomic classification were performed utilising the HMMER 3.2.1 platform [20] in conjunction with the CAZy database (Carbohydrate-Active Enzymes Database; http://bcb.unl.edu/dbCAN2, accessed on 13 September 2024) [21], applying stringent filtering criteria (E-value cut-off < 1 × 10^−5^; sequence coverage > 35%). This analytical framework effectively integrates the Hidden Markov Model (HMM) algorithm, thereby ensuring the biological robustness of the annotation results.

### 2.4. SNP Site Detection

The present study established a systematic genomic data processing workflow using the Illumina paired-end sequencing raw data (FASTQ format) and the corresponding genome assembly file of *G. lapidescens* QL01 as primary analytical inputs. Initially, the Burrows-Wheeler Transform (BWT) algorithm was employed, specifically the FM-index method [22]. The BWA v0.7.17 software’s index function was utilised to construct a sequence index of the reference genome, generating standard index files (including .amb, .ann, .bwt, .pac, and .fai formats). Subsequently, the BWA-MEM algorithm [22] was applied to perform high-sensitivity alignment of the paired-end sequencing reads to the reference genome, yielding sequence alignment maps in SAM format.

The primary data processing, incorporating such functions as format conversion and quality filtering, was executed utilising the SAMtools v1.13 software. Initially, the SAM file was converted into the more efficient binary BAM format utilising the view command. Subsequently, the reads were filtered to retain high-quality alignments, employing a stringent criterion of Q ≥ 30. Subsequently, the faidx command within SAMtools was employed to generate a genome index file, thereby ensuring coordinate system consistency and compatibility for downstream analyses. Subsequent to the initial processing, standardised pre-processing was conducted using the Genome Analysis Toolkit (GATK) v4.3.0.0 [23]. The BAM file was sorted using the SortSam module to optimise data structure. Subsequently, the MarkDuplicates module was implemented to identify and eliminate PCR duplicate sequences. Concurrently, the integrity and validity of the index files were validated to ensure data accuracy and reliability.

During SNP detection phase, variant calling across the genome was performed using the GATK Haplotype Caller module based on a haplotype-based assembly strategy. The generation of Genome Variant Call Format (gVCF) files was subsequently facilitated, with population-level variant sites then being integrated via the Genotype GVCFs module. The filtered SNP dataset was subsequently converted to MAP format using PLINK v1.9 [24], facilitating multi-dimensional data visualisation through an in-house developed Python V 3.9 tool.

### 2.5. Reconstruction of the Phylogenetic Tree

Orthologous genes in *G. lapidescens* QL01, 30 representative basidiomycetes, and a single ascomycete outgroup were identified using OrthoFinder v2.5.1 [25]. Protein sequence alignments of single-copy orthologues were performed with MAFFT v7.205 [26] under default parameters. The annotation and characterisation of unique gene families was achieved through Pfam database screening. Gene Ontology (GO) and KEGG enrichment analyses were conducted employing clusterProfiler v3.6.0. The optimal substitution model was determined via the implementation of Model Finder in IQ-TREE v1.6.11 [27]. A maximum likelihood phylogeny was subsequently constructed using RAxML with the best-fit substitution model, supported by 1000 bootstrap replicates. The estimation of divergence times was conducted using the MCMCTree programme in PAML v4.9i [28], incorporating two calibration points obtained from the TimeTree database [29]. The following comparisons are made between *Dacryopinax primogenitus* and *Calocera cornea* (76.8–94.1 MYAs) and *Acaromyces ingoldii* and *Ustilago maydis* (211.4–370.0 MYAs). The time-calibrated phylogeny was subsequently visualised using MCMCTree [28].

### 2.6. Gene Family Expansion and Contraction Analysis

The expansion and contraction of gene families were investigated using CAFE 5.0 [30], integrating the dataset of identified gene families, the reconstructed phylogenetic tree, and estimated divergence times. Within the CAFE framework, a stochastic birth and death model was employed to trace evolutionary trajectories of gene gains and losses across the phylogenetic tree. For each gene family, conditional *p*-values were computed, with values below 0.05 deemed indicative of a significant shift in the rate of gene gain or loss. The analytical results were subsequently visualised graphically using iTOL [31], providing an intuitive representation of the evolutionary dynamics of gene families within the studied taxa.

### 2.7. BGC Analysis and Visualisation

The study employed an integrated multi-omics approach for the systematic analysis of Biosynthetic Gene Clusters (BGCs). Initially, genomic localisation and functional annotation of BGCs were performed utilising the antiSMASH 7.0 platform [32]. Subsequently, the evolutionary relationships of the BGCs were elucidated by conducting phylogenetic reconstruction of the BGCs using IQ-TREE 2.2.3 software [27]. The algorithm parameters were specified as -m MFP -bb 1000 -alrt 1000 -abayes -nt AUTO with a view to enhancing computational efficiency. The investigation focused on key multi-domain enzymes, including nonribosomal peptide synthetases (NRPSs), NRPS-like enzymes, ribosomally synthesised and post-translationally modified peptides (RiPPs), and synthases (as per Synthaser) [33]. An in-depth domain architecture analysis of these enzymes was performed via the antiSMASH platform, with the aim of identifying critical functional modules such as adenylation (A), thiolation (T), thioesterase (TE), condensation (C), and thioester reductase (TR) domains.

### 2.8. Partial Core Gene Methylation Prediction

The advent of contemporary long-read sequencing technologies has had a considerable impact on the streamlining of library preparation workflows. Third-generation sequencing platforms, exemplified by PacBio Single Molecule, Real-Time (SMRT) sequencing and Oxford Nanopore Technologies (ONT) nanopore sequencing, have been shown to simultaneously deliver long-read sequences and enable the detection of DNA methylation modifications [34]. The most recent high-quality PacBio HiFi data combines long-read lengths with high accuracy, permitting the direct detection of DNA 5-methylcytosine (m5C). This methylation information is encoded within the raw signal data (fluorescence pulses for PacBio). The specific methylation sites can be determined by analysing temporal delays in the fluorescence kinetics (PacBio SMRT sequencing).

Contemporary long-read sequencing technologies offer streamlined library preparation workflows. Third-generation sequencing platforms, exemplified by PacBio Single Molecule, Real-Time (SMRT) sequencing and Oxford Nanopore Technologies (ONT) nanopore sequencing, have the capacity to simultaneously generate long reads and enable the detection of DNA methylation modifications. The most recent high-fidelity PacBio HiFi data combines long-read lengths with high accuracy, permitting direct detection of m5C. This methylation information is encoded within the raw pulse signals of the primary sequencing data, where specific methylation sites can be determined by analysing inter-pulse duration (IPD) deviations. The primary processing was conducted utilising SMRT Link v13.1. Within this platform, m5C was predicted employing the integrated jasmine algorithm. Subsequently, the processed reads were aligned to the reference genome using pbalign, yielding a tagged BAM file. Subsequent methylation analysis on the aligned BAM file was conducted utilising pb-CpG-tools (https://github.com/PacificBiosciences/pb-CpG-tools, accessed on 13 May 2025), producing a methylation-annotated BAM file. Finally, the visualisation process was executed through the utilisation of bespoke Python scripts.

The processing and analysis of the HIFI data was conducted using PacBio’s official SMRT Link v13.1 software suite, with integrated prediction of CpG and 5-methylcytosine (5mC) sites performed via the jasmine module. Subsequent to this, the processed reads were aligned to the reference genome employing minimap2 [35], yielding an indexed BAM file. Subsequently, a methylation profiling procedure was executed using pb-CpG-tools, thereby generating methylation-annotated BAM files. The process of visualisation was executed through the utilisation of bespoke Python scripts (Appendix A).

### 2.9. Data Availability

The ITS sequence of *G. lapidescens* QL01 was registered in the NCBI GenBank under accession number PX125820, and the final genome assembly and associated datasets for *G*. *lapidescens* QL01 have been submitted to NCBI under BioProject PRJNA 1256626, and BioSample SAMN 48188347.

## 3. Results

### 3.1. Collection and Domestication Cultivation of Wild Macrofungi Resources in the Qinling Mountains

Recent years have seen the Qinling *G. lapidescens* industry achieve significant developmental progress, establishing itself as a notable emerging segment within the broader context of China’s TCM sector. The Qinling region is of particular interest in the study of wild medicinal fungi, due to its status as a globally significant repository. One notable species is *G. lapidescens*, which is distinguished by its distinctive medicinal properties [5,6]. In the context of scientific research focused on the exploration and utilisation of wild macrofungal resources in the Qinling Mountains, a significant number of freshly harvested, viable *G. lapidescens* specimens were obtained. Through rigorous strain isolation and domestication cultivation procedures from these samples, a distinct strain was identified and characterised. This strain was designated *G. lapidescens* QL01. This particular isolate displays unique mycelial morphological characteristics, indicating a substantial deviation from the characteristics of commercially available cultivars. A thorough evaluation has been conducted, which has confirmed the strain’s superior and stable medicinal attributes. This has led to its formal designation as *G. lapidescens* QL01. In view of the pressing need for innovative research on core germplasm resources in the Qinling region, which is the emerging production area of *G. lapidescens*, and the current dearth of genomic research on the species, it is imperative to undertake genomic research on this exceptional wild strain. This initiative is imperative in order to address a significant gap in academic knowledge and to promote industrial development.

### 3.2. Genome Sequencing, De Novo Assembly, and Annotation

The genome size of *G. lapidescens* QL01 was estimated at approximately 82.23 Mbp through K-mer analysis (Appendix A), accompanied by a heterozygosity rate of 0.52%, a repetitive sequence proportion of 35.51%, and a guanine-cytosine (GC) content of approximately 45.0% (Appendix A). The strain’s genome was assembled utilising a multi-platform sequencing strategy. This integrated 6.98 Gbp of BGI short-read sequencing data (~99.98% coverage) and 4.74 Gbp of HiFi long-read data (~99.93% coverage) (Appendix A), yielding a final assembly of 82.23 Mbp (Figure 1). The initial assembly performed with HiFiSM resulted in a contig-level genome totalling 82.23 Mbp. This assembly was subsequently anchored to 11 chromosomes, alongside 12 unscaffolded contigs (Appendix A). The contig N50 was found to be 7.15 Mbp, with a GC content of 46.03%. The range of chromosome lengths observed was from 1,713,274 bp to 7,185,557 bp, as illustrated in Figure 1 and detailed in Appendix A. Assembly quality metrics indicated high completeness: an average depth of coverage of 254.88×, a genome coverage of 99.91%, and a BUSCO completeness score of 98.4%. Specifically, BGI sequencing achieved an average depth of 84.9× and 99.91% coverage, while HiFi sequencing yielded 57.6× depth and 99.91% coverage. Concurrently, BUSCO analysis revealed 98.4% genome completeness, with 96.2% of BUSCO genes present as single copies (Appendix A). Collectively, these data demonstrate the high quality and completeness of the genome assembly.

The genome assembly of *G. lapidescens* QL01 comprises 15,847 predicted protein-coding genes. The mean transcript length was found to be 2361.46 bp, with an average coding sequence (CDS) length of 1498.04 bp (Appendix A). Assessment using BUSCO (fungi_odb10 database) indicated that 93.4% of genes were complete (comprising 91.0% single-copy and 2.4% duplicated genes), while 1.2% were fragmented and 5.4% were missing (Appendix A). Collectively, these metrics demonstrate high completeness and accuracy in gene prediction. Functional annotation was performed against five databases: The following databases were consulted: SwissProt, NR, Pfam, Gene Ontology (GO) and KEGG. In total, 12,855 genes (81.12%) were functionally annotated in at least one database. Annotation rates exhibited significant variation, with the highest rates observed in the NR database (11,135 genes; 86.62%), and the lowest rates observed for KEGG Orthology (KO) annotations (3484 genes; 27.10%) (Appendix A).

### 3.3. SNP and Methylation Analysis of Gerronema lapidescens QL01

Genome-wide polymorphism analysis is fundamental for functional gene mapping and studies of genetic diversity. A thoroughgoing investigation of Illumina NovaSeq sequencing data has led to the identification of 8,460,308 high-confidence SNPs within the *G. lapidescens* QL01 genome, distributed across all chromosomes (Figure 2A, Appendix A). Concurrently, methylation analysis revealed 3,246,792 methylation events across the 11 chromosomes of *G. lapidescens* QL01, representing the methylation density per 100 kilobases (per 100 kb). It is hypothesised that DNA methylation may induce gene silencing by impeding transcription factor binding. Transcription factors generally necessitate the recognition of specific DNA sequences to initiate transcription; the steric hindrance imposed by methyl groups has been shown to physically obstruct this binding, thereby preventing transcriptional initiation and leading to gene silencing. Consequently, elevated methylation frequencies were observed on Chr4, Chr9, and Chr10, suggesting a potential association with increased gene silencing in these regions. Conversely, Chr6 exhibited the lowest frequency of methylation events (Figure 2B, Appendix A), indicating comparatively higher evolutionary conservation within this chromosome during the species’ metabolic evolution relative to others.

### 3.4. Phylogenetic and Gene Family Variation Analysis

To clarify the phylogenetic position and divergence times of *G. lapidescens* QL01, this study constructed a phylogenetic tree of representative parasitic and basidiomycete species, using Ustilago maydis (Ascomycota) as the outgroup. Within the Basidiomycota phylum, *G. lapidescens* QL01 showed the closest evolutionary relationship to *L. edodes* and *D. bispora*, diverging approximately 130.883 MYAs, with a 95% HPD interval of 3.253 to 6.178 million years (Figure 3, Appendix A). This divergence likely resulted from distinct habitat adaptations: *G. lapidescens* QL01 grows on rock surfaces under extreme conditions, unlike *L. edodes* and *D. bispora*, which typically grow on wood or decaying plant matter, reflecting their evolutionary adaptation to different environments. Further analysis of the reconstructed phylogenetic tree revealed complex patterns of gene family expansion and contraction among 66,178 gene families across 31 species. In *G. lapidescens* QL01, 301 of the 1400 gene families underwent expansion or contraction. Compared to *L. edodes* (329 gene families) and *D. bispora* (929 gene families), *G. lapidescens* QL01 experienced relatively minor gene family changes. This suggests that its gene family dynamics might reflect adaptations to a rock-based environment. These gene family alterations likely provided *G. lapidescens* QL01 with significant evolutionary advantages for survival and reproduction in variable environments.

### 3.5. CAZyme Analysis and Developing SSR Markers

*Gerronema lapidescens* QL01 has been observed to thrive in damp, shady environments, typically growing on rock surfaces or in stony soil. It is characterised by its strong adaptability and stress tolerance, which enable it to degrade organic matter in rocks, releasing nutrients into the environment. Furthermore, it plays a pivotal role in the processes of rock weathering and soil formation. The strain under scrutiny contains 521 CAZymes, including 211 glycoside hydrolases (GHs), 62 glycosyltransferases (GTs), 162 auxiliary activities (AAs), 27 carbohydrate esterases (CEs), 12 polysaccharide lyases (PLs), and 47 carbohydrate-binding modules (CBMs) (Figure 4A, Appendix A). Correlation analysis indicates a close relationship between the GH proportion in 211 enzymes of *G. lapidescens* QL01 and Stropharia rugosoannulata and their living conditions. In terms of CAZyme characteristics, *G. lapidescens* is most similar to *Cyclocybe aegerita* (see Figure 4A). The application of cluster analysis has resulted in the grouping of *G. lapidescens* and *C. aegerita* with certain members of the *Lentinula edodes* species (Appendix A).

In the context of these fungi, simple sequence repeats (SSRs) have been observed to be distributed extensively throughout both coding and non-coding regions of genes, thereby playing a pivotal role in the regulation of gene expression and the evolution of species. Due to their high degree of polymorphism and extensive genomic distribution, they are frequently regarded as ideal molecular markers [36]. A genome-wide scan of the *G. lapidescens* QL01 sequence (Figure 4B and Appendix A) revealed 3279 SSRs with motif unit lengths ranging from 1 to 6 nucleotides, at a relative frequency of approximately 155 SSRs per million base pairs in the *G. lapidescens* QL01 genome (56.04 million base pairs). Intriguingly, in the SSR composition of *G. lapidescens* QL01, the (TAT)52 repeat sequence accounted for the highest proportion at 46.39%. A comparative SSR analysis of four *related macrofungi*, including *Wolfiporia cocos*, was conducted, which revealed that trinucleotide repeats are a major component of SSRs in these species, with all exceeding 40% (Appendix A).

### 3.6. Search and Analysis of Secondary Metabolite-Related Genes

In view of the substantial medicinal value of *G. lapidescens* QL01, notably its anthelmintic properties for paediatric malnutrition and antitumour activities, a comprehensive search and analysis of secondary metabolite biosynthesis gene clusters within its genome was undertaken. The genome was interrogated using AntiSMASH (Figure 5A and Table 1). The prediction results indicate that the genome of *G. lapidescens* QL01 contains 67 gene clusters, which collectively encode 76 core genes. These include 26 enzymes associated with terpene synthesis, 21 enzymes similar to NRPSs, 18 enzymes similar to RiPPs, seven enzymes belonging to type I polyketide synthase, three enzymes similar to NRPSs, and one enzyme similar to an iron-siderophore transporter. The distribution of these BGCs across the 11 chromosomes is as follows: Chr11 harbours the highest concentration (11 core genes), while Chr9 harbours the lowest (2 core genes).

As core genes play a crucial role in secondary metabolite synthesis, the relevant genes were subjected to further analysis. Sesquiterpenes are a class of major bioactive components found in large fungi; as such, the focus of this study was on the terpene synthases annotated in the strain *G. lapidescens* QL01. In accordance with the predictions made by AntiSMASH, a total of 19 monoterpene synthases were selected from the 26 terpene synthases that were predicted (Appendix A). The 19 STSs from *G. lapidescens* QL01 were divided into four groups. The sesquiterpene synthase group was characterised by 1,11-cyclisation of (2E,6E)-FPP, consisting of eight enzymes (Figure 5B). AntiSMASH predictions revealed that *G. lapidescens* QL01 contains 31 NRP, NRPS-like and PKS-encoding genes. A multidomain analysis of these 31 genes revealed that Gsp01766 and Gsp12097 contain multiple C, A, and T domains, suggesting that these genes may encode cyclodimer formation (Figure 5C). It is notable that the NRPS-like associated encoding genes in question generally contain A, T, TR domains; however, Gsp06292, Gsp07102, Gsp07094, and Gsp11070 are notable exceptions in that they lack the T domain. It is hypothesised that genes devoid of the T domain may be implicated in disparate metabolic branches, or alternatively, their functions within metabolic pathways may have undergone modification. Furthermore, the existence of a variety of NRP, NRPS-like and PKS-like encoding genes in *G. lapidescens* QL01 facilitates the estimation of the diversity of non-ribosomal peptide compounds in this strain.

### 3.7. Methylation Analysis of the BGCs for Secondary Metabolite

Gene methylation-based silencing, a pivotal epigenetic regulatory mechanism, functions by suppressing gene expression and maintaining genomic stability through the addition of methyl groups to specific DNA bases. In this study, we identified methylated gene clusters and utilised a Python script to visualise the methylation data, thereby linking methylation sites to core genes (Figure 6, Appendix A). Predictions indicate that Gsp01766 in Cluster 13 exhibits the highest degree of methylation, suggesting the possibility of complete silencing and subsequent non-functional status. Furthermore, core genes in Clusters 21, 29, and 60 (Gsp02170, Gsp04909, Gsp12017) demonstrate high methylation, suggesting the possibility of silencing. Conversely, core genes within Clusters 38, 42, 50, and 53 (Gsp06593, Gsp07849, Gsp09146, Gsp10648) exhibited a paucity of methylation sites, suggesting a reduced probability of silencing.

## 4. Discussion

The Qinling Mountains represent a significant botanical demarcation line between northern and southern China, exhibiting characteristics of both subtropical and temperate climates. The region is distinguished by its abundant forestation and varied flora, which collectively foster a distinctive forest ecosystem. This region is distinguished by its rich macrofungal resources, comprising a considerable number of documented species. The Qinling Mountains have gained international renown as a natural repository of medicinal substances, with a particular abundance of medicinal fungi. In the TCM domain, a number of species are of significant medicinal value. These include *Cordyceps* spp., *Polyporus umbellatus*, and *Boletus* spp., which are extensively utilised. It is noteworthy that several novel macrofungal species have been discovered in the Qinling region in recent years [37,38]. The primary objective of the present research group is the exploration of the germplasm resources of macrofungi located within the geographical confines of the Qinling Mountains. Utilising tissue isolation techniques, we have successfully isolated and cultivated numerous saprophytic macrofungi, including *Laetiporus sulphureus* [39], *Cryptoporus qinlingensis* [40], *Cyathus olla* [41], and *Hericium rajendrae* [42]. Among these, fungi possessing both edible and medicinal value, such as *Cryptoporus qinlingensis* [40] and *Hericium rajendrae* [42], are considered particularly valuable. The Qinling Mountains constitute a natural boundary between the central and southern regions of Shaanxi Province. Moreover, these regions represent the primary edible mushroom production areas in southern Shaanxi. Sclerotium-forming fungi represent a category of fungi capable of producing sclerotia. These are compact, dormant masses of fungal mycelium (propagules) formed under specific conditions, exhibiting strong regenerative capacity [43]. In TCM, the term “Three Ling” is used collectively to describe three specific types of fungi: *Omphalia lapidescens*, *Polyporus umbellatus*, and *Wolfiporia extensa*. All of these organisms are sclerotium-forming fungi, and their sclerotia are utilised in the context of medicine. While they do share certain therapeutic properties, such as promoting diuresis and reducing oedema, each possesses distinct efficacies and applications [44]. The *G. lapidescens* QL01 strain utilised in this study was meticulously collected from the core area of the Qinling Mountains. It is hypothesised that the findings from this research will provide essential data to support the in-depth development and rational utilisation of *G. lapidescens* resources within the Qinling region.

In this study, the first HIFI genome assembly for the *G. lapidescens* QL01 strain was achieved, with the genome successfully anchored onto 11 chromosomes. The final genome size was determined to be 87.49 Mbp, with an N50 length of 7.15 Mbp and a BUSCO completeness score of 98.4%. The results of this study address, to a certain extent, the paucity of high-quality genomic resources available within the *Gerronema* genus, thereby establishing a robust foundation for future research on this taxon. A comparative genomic analysis was conducted between *G. lapidescens* QL01 and other basidiomycetes. This analysis revealed its closest evolutionary affinities to *L. edodes* and *D. bispora*. The divergence time between *G. lapidescens* and these two species is estimated at 130.883 MYAs, with a 95% HPD interval of 3.253 to 6.178 million years. It is hypothesised that this divergence is the result of distinct habitat adaptations. A comparative analysis of CAZymes across 16 basidiomycetes, including *G. lapidescens* QL01, revealed a strong correlation specifically between *G. lapidescens* QL01 and Stropharia rugosoannulata strains. A notable finding was the observation of a higher proportion of GHs among the 211 shared enzymes in these correlated species, which suggests a potential link to their ecological niches or life strategies. SNP and methylation are two distinct types of variation commonly found at different levels within a genome. This study demonstrates that the *G. lapidescens* genome exhibits extensive variation in both types, which may be associated with its adaptation to the complex ecological environment of the Qinling Mountains. SNP analysis of the shiitake mushroom population has revealed potential patterns of variation [45], whereas the methylation level in the *Heterobasidion parviporumis* genome is closely related to its survival strategy [46]. It should be emphasised that the variation reflected by SNPs was called from a single sequenced strain QL01, and therefore reflects heterozygosity within the genome rather than the population-level diversity of *G. lapidescens*. Moreover, this study constitutes the initial comprehensive examination of SSRs within the *G. lapidescens* species. The identification of polymorphic SSRs across different strains provides essential groundwork for the development of genetic molecular markers.

Macrofungi are frequently a significant source of bioactive constituents [47,48,49,50,51], and sclerotium-forming represent a distinctive category within the macrofungal kingdom. Sclerotium-forming possess distinctive biological properties and a plethora of secondary metabolites, which render them highly promising candidates for medicinal applications [43]. The sclerotia of renowned species such as *Polyporus umbellatus* (Zhuling), *Wolfiporia extensa* (Fuling), and *G. lapidescens* (Lei Wan) have long been held in high esteem in TCM, where they are employed as therapeutic agents for the prevention and treatment of various ailments [4]. The nutritional and bioactive properties of the subject are intrinsically linked to the secondary metabolic pathways of species such as *G. lapidescens* QL01. For instance, analysis of the *G. lapidescens* QL01 secondary metabolome using antiSMASH led to the identification of 67 distinct BGCs. Core gene analysis revealed a predominance of terpene synthases, indicating a substantial potential for the synthesis of diverse terpenoid compounds within this strain. This finding provides crucial theoretical guidance for the future development of bioactive compounds from *G. lapidescens* QL01. Concurrently, the research team identified BGCs harbouring predicted methylation modifications. Utilising bespoke Python scripts, we conducted a visualisation analysis of the methylation data, mapping core methylation sites onto their corresponding core genes. Among these, the core gene Gsp01766 exhibited the highest degree of methylation, thus suggesting the potential for complete silencing of this gene. Furthermore, natural product chemistry analysis of the secondary metabolites of *G. lapidescens* fails to match the diverse biosynthetic potential demonstrated by its BGCs. The most likely explanation is that methylation silencing renders some BGCs incapable of synthesising compounds.

## 5. Conclusions

This study presents the first genome sequencing and high-quality assembly of a *G. lapidescens* QL01 strain originating from the Qinling region. The assembled genome of *G. lapidescens* QL01 spans 87.49 Mbp and was successfully anchored onto 11 chromosomes utilising HiFi technology. A series of comparative analyses were conducted, incorporating methylation profiling, classification and quantification of CAZymes, and investigations into gene family expansion and contraction. These analyses indicated a divergence time of approximately 130.883 MYAs between this strain and its closest relatives. This divergence may be attributable to distinct adaptations to specific habitats. A detailed analysis of secondary metabolite biosynthesis was conducted, which revealed that the *G. lapidescens* QL01 genome encodes 67 BGCs, comprising 76 core genes. The methylation prediction of core genes identified Gsp01766 as exhibiting the highest methylation level, suggesting the potential for complete silencing of this gene. Collectively, this research provides a high-quality genomic resource for *G. lapidescens* QL01, thereby establishing a solid foundation for understanding its genetic evolution and facilitating biotechnological applications such as molecular breeding.

## Figures and Tables

**Figure 1 jof-11-00647-f001:**
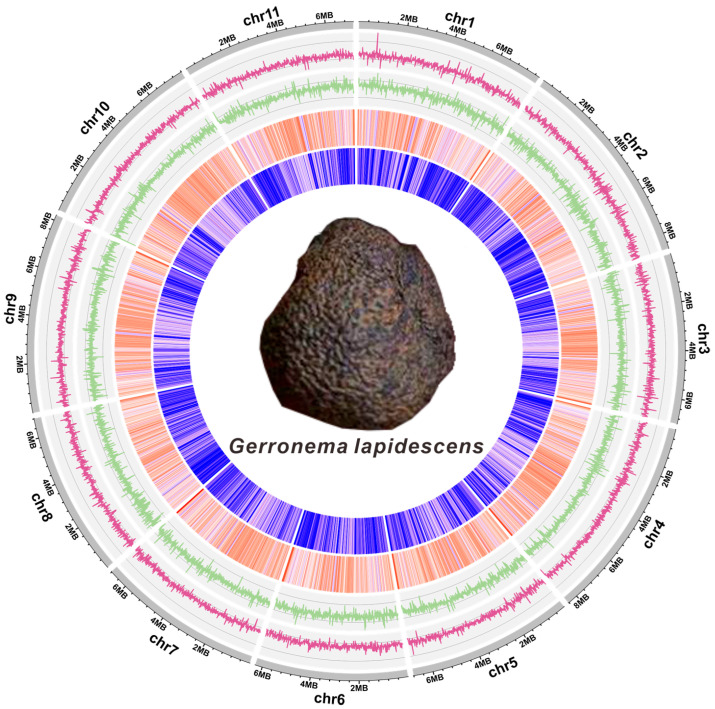
Genomic features of *Gerronema lapidescens* QL01. The concentric tracks, from the outermost to the innermost, I. chromosomes and contigs, II–IV. GC content, GC skew and AT skew (1 kb window size), V. Gene density (1 kb window size). A photograph of the sclerotial morphology of *Gerronema lapidescens* is shown in the centre.

**Figure 2 jof-11-00647-f002:**
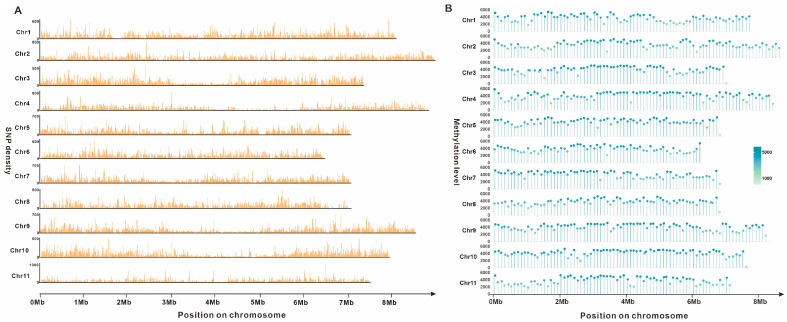
The SNP (window size 10 kb, (**A**)) and methylation (window size 100 kb, (**B**)) analyses of *Gerronema lapidescens* QL01.

**Figure 3 jof-11-00647-f003:**
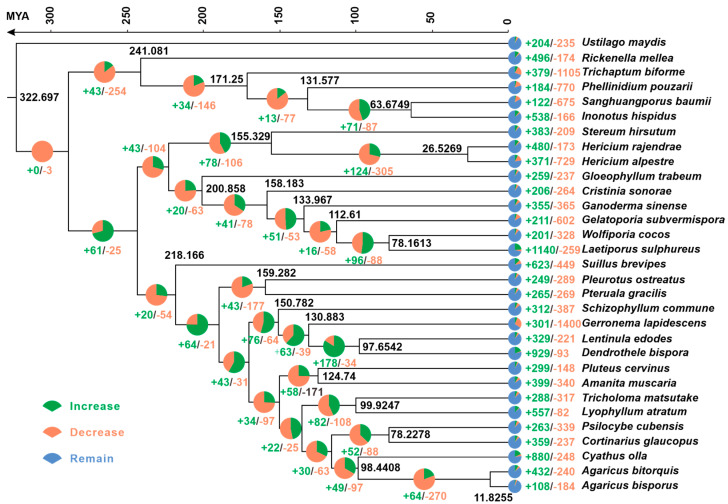
Phylogeny and gene family variation. The phylogenetic tree depicting the evolutionary relationships of *Gerronema lapidescens* QL01 with 30 representative basidiomycete species. Black numerals on branches indicate the crown-group mean age (MYA) for each node. The bar chart on the right illustrates the number of significantly expanded (green) and contracted (orange) gene families in each species.

**Figure 4 jof-11-00647-f004:**
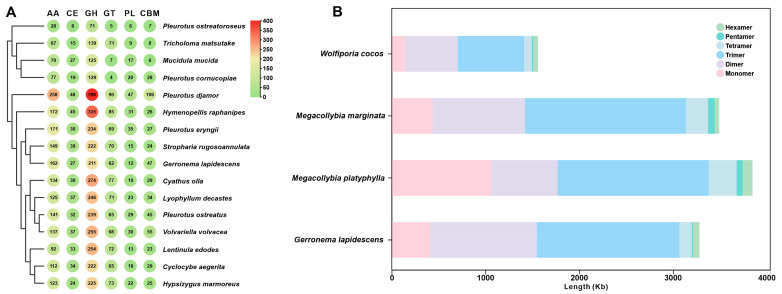
CAZyme (**A**) and SSR (**B**) analyses of *Gerronema lapidescens* QL01 and related macrofungi.

**Figure 5 jof-11-00647-f005:**
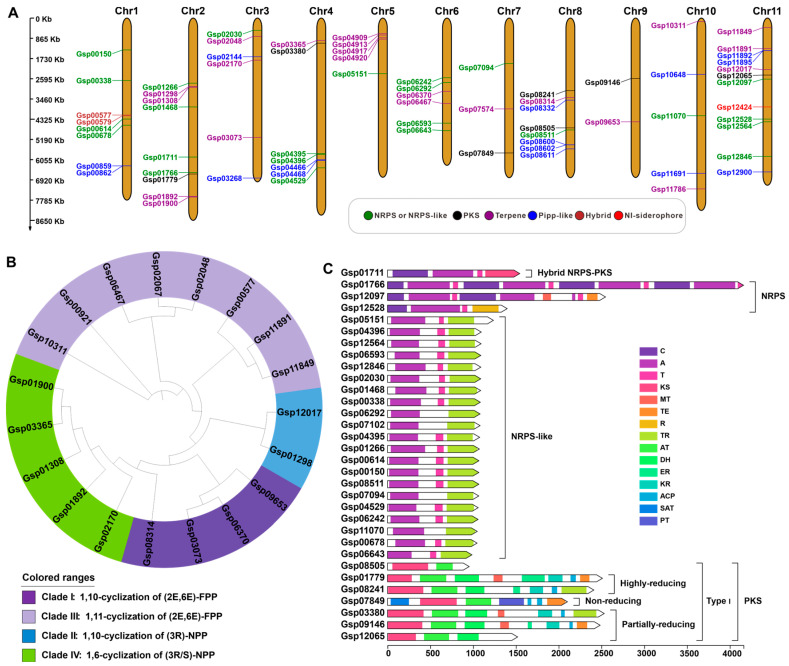
Analysis and characterisation of core genes related to secondary metabolite biosynthesis in the *Gerronema lapidescens* QL01 Genome. (**A**) The core genes distribution of BGCs on chromosomes. (**B**) Phylogenetic tree analysis of STSs. (**C**) The structural characteristics of the enzymes with multiple domains.

**Figure 6 jof-11-00647-f006:**
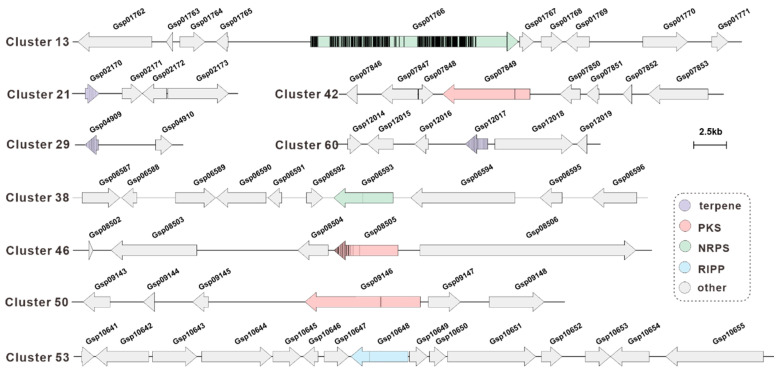
The methylation prediction of select core genes in *Gerronema lapidescens* QL01. The representation of a methylation site is indicated by a black line.

**Table 1 jof-11-00647-t001:** Putative BGCs for secondary metabolites in the genome of *G. lapidescens* QL01.

Cluster. No	Location			Core Gene ID	Core Gene Type
	Chromosome	Start (bp)	End (bp)		
1	Chr1	1,333,163	1,362,998	geneGsp00150	NRPS-like
2	Chr1	2,660,560	2,701,903	geneGsp00338	NRPS-like
3 *	Chr1	4,134,889	4,192,917	geneGsp00577	Terpene
				geneGsp00579	Ripp-like
4	Chr1	4,285,042	4,327,761	geneGsp00614	NRPS-like
5	Chr1	4,548,301	4,587,172	geneGsp00678	NRPS-like
6 *	Chr1	6,317,779	6,318,426	geneGsp00859	Ripp-like
				geneGsp00862	Ripp-like
7	Chr1	6,967,152	6,971,998	geneGsp00921	Terpene
8	Chr2	2,763,751	2,792,130	geneGsp01266	Terpene
9	Chr2	2,908,572	2,926,762	geneGsp01298	Terpene
10	Chr2	2,936,974	2,952,672	geneGsp01308	Terpene
11	Chr2	3,789,149	3,818,583	geneGsp01468	NRPS-like
12	Chr2	5,934,048	5,954,781	geneGsp01711	NRPS
13	Chr2	6,591,072	6,638,658	geneGsp01766	NRPS
14	Chr2	6,650,704	6,697,700	geneGsp01779	T1PKS
15	Chr2	7,615,453	7,622,734	geneGsp01892	Terpene
16	Chr2	7,650,053	7,662,432	geneGsp01900	Terpene
17	Chr3	505,392	528,870	geneGsp02030	NRPS-like
18	Chr3	760,222	770,517	geneGsp02048	Terpene
19	Chr3	1,026,023	1,044,018	geneGsp02067	Terpene
20 *	Chr3	1,645,178	1,724,472	geneGsp02144	Ripp-like
				geneGsp02153	Ripp-like
21	Chr3	1,803,842	1,814,061	geneGsp02170	Terpene
22	Chr3	5,087,409	5,105,567	geneGsp03073	Terpene
23	Chr3	6,804,933	6,858,659	geneGsp03268	Ripp-like
24	Chr4	954,860	965,271	geneGsp03365	Terpene
25	Chr4	1,064,339	1,106,327	geneGsp03380	T1PKS
26 *	Chr4	5,766,937	5,836,976	geneGsp04395	NRPS-like
				geneGsp04396	NRPS-like
27 *	Chr4	6,042,896	6,113,623	geneGsp04466	Ripp-like
				geneGsp04468	Ripp-like
28	Chr4	6,394,666	6,423,026	geneGsp04529	NRPS-like
29	Chr5	674,185	680,885	geneGsp04909	Terpene
30	Chr5	721,838	723,100	geneGsp04913	Terpene
31	Chr5	826,981	832,702	geneGsp04917	Terpene
32	Chr5	897,400	908,671	geneGsp04920	Terpene
33	Chr5	2,345,287	2,387,459	geneGsp05151	NRPS-like
34	Chr6	2,525,938	2,563,354	geneGsp06242	NRPS-like
35	Chr6	2,721,335	2,761,718	geneGsp06292	NRPS-like
36	Chr6	3,126,334	3,144,062	geneGsp06370	Terpene
37	Chr6	3,648,047	3,664,942	geneGsp06467	Terpene
38	Chr6	4,484,271	4,526,656	geneGsp06593	NRPS-like
39	Chr6	4,802,482	4,842,126	geneGsp06643	NRPS-like
40 *	Chr7	1,909,694	1,988,097	geneGsp07094	NRPS-like
				geneGsp07102	NRPS-like
41	Chr7	3,890,379	3,906,718	geneGsp07574	Terpene
42	Chr7	5,762,576	5,790,301	geneGsp07849	T1PKS
43	Chr8	3,083,461	3,128,991	geneGsp08241	T1PKS
44	Chr8	3,391,092	3,400,084	geneGsp08314	Terpene
45	Chr8	3,505,329	3,516,333	geneGsp08332	Ripp-like
46	Chr8	4,667,825	4,708,615	geneGsp08505	T1PKS
47	Chr8	4,749,215	4,786,027	geneGsp08511	NRPS-like
48 *	Chr8	5,397,733	5,458,632	geneGsp08600	Ripp-like
				geneGsp08602	Ripp-like
49 *	Chr8	5,581,739	5,639,225	geneGsp08611	Ripp-like
				geneGsp08613	Ripp-like
50	Chr9	2,552,571	2,584,816	geneGsp09146	T1PKS
51	Chr9	4,433,774	4,453,262	geneGsp09653	Terpene
52	Chr10	122,324	141,261	geneGsp10311	Terpene
53	Chr10	2,391,981	2,444,186	geneGsp10648	Ripp-like
54	Chr10	4,164,792	4,201,924	geneGsp11070	NRPS-like
55	Chr10	6,619,032	6,669,729	geneGsp11691	Ripp-like
56	Chr10	7,300,583	7,307,881	geneGsp11786	Terpene
57	Chr11	396,663	397,903	geneGsp11849	Terpene
58	Chr11	1,307,102	1,308,342	geneGsp11891	Terpene
59 *	Chr11	1,386,710	1,394,778	geneGsp11892	Ripp-like
				geneGsp11895	Ripp-like
60	Chr11	2,201,870	2,220,088	geneGsp12017	Terpene
61	Chr11	2,403,505	2,440,506	geneGsp12065	T1PKS
62	Chr11	2,594,189	2,636,978	geneGsp12097	NRPS
63	Chr11	3,787,596	3,815,670	geneGsp12424	NI-siderophore
64	Chr11	4,301,554	4,334,404	geneGsp12528	NRPS-like
65	Chr11	4,411,006	4,449,199	geneGsp12564	NRPS-like
66	Chr11	5,908,891	5,949,936	geneGsp12846	NRPS-like
67	Chr11	6,551,530	6,585,327	geneGsp12900	Ripp-like

* indicates that the cluster contains more than one core gene.

## Data Availability

The original contributions presented in this study are included in the article/Appendix A. Further inquiries can be directed to the corresponding authors.

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
