# Peer review of "The First Whole Genome Sequence and Methylation Profile of Gerronema lapidescens QL01"

_jof, 2025, doi:10.3390/jof11090647_

Round 1
Reviewer 1 Report
Dear authors,
Your article “The First Whole Genome Sequence and Methylation Profile of Gerronema lapidescens QL01” undoubtedly demonstrates a high degree of novelty, as the manuscript presents, for the first time, the genome sequencing and assembly of the G. lapidescens QL01 strain. However, while reading your article, I had a number of questions and comments.
Dear authors, unfortunately, the version of the manuscript that was provided to me for review lacks line numbering, which somewhat complicates the review process.
- The purpose of the research is not stated in the manuscript.
- Page 2 – “Bioactive compounds include G. lapidescens lectin and sterols [7,8], conferring antitumour, antiparasitic, anthelmintic, anti-inflammatory, antiviral, immunomodulatory, and hypoglycaemic activities[9]” – you are referring to a non-existent article published in a non-existent journal. [7] - Yonghai, Y.; Jun, G.; Mingkun, Y. Studies on the purification and properties of Omphalia lapidescens lectin. Systematics of Mycology 2000, 278-282, doi:10.13346/j.mycosystema.2000.02.023 . Did you use generative neural networks when writing the article?
- In the Materials and Methods section, there is no information about the source of the lapidescens QL01 strain used in the research, and there is also no characteristic of the organism.
- Page 2, Section “2.1 Fungal material and nucleic acid extraction” - “The high-quality mycelia of G. lapidescens QL01 were cultured in PDB liquid medium (25°C, 180 rpm, one week) to obtain mycelium for DNA extraction.” - It is necessary to specify the composition of the PDB nutrient medium or provide a link to the article where it is indicated.
- Section “2.1 Fungal material and nucleic acid extraction” – Was there any mycelium treatment before using the Fungal Genomic DNA Rapid Extraction Kit (Sangon Biotech, Shanghai, China)?
- In the Materials and Methods section, when mentioning the devices that were used for this or that manipulation, please write the company and the country of the manufacturer.
- Page 3 – “…and scaffolds evaluated for continuity, accuracy, consistency, and completeness.” -how was this check conducted? You need to add a reference.
- Page 3, Section “2.4 SNP site detection” - “Initially, the Bur-rows-Wheeler Transform (BWT) algorithm was employed, specifically the FM-index method” – you need to add references.
- Page 5, fourth paragraph – “The process of visualization was executed through the utilization of bespoke Python scripts.” – did you use standard scripts or copyrighted ones? If these were author's scripts, then you must provide a link to the repository with the code or the script attachment to the article.
- Section “2.9. Data Availability” – “The final genome assembly and associated datasets for G. lapidescens QL01 have been submitted to NCBI under BioProject PRJNA 1256626, and BioSample SAMN 48188347.” – Using the specified genome assembly data, I was unable to view the genome in the NCBI database. I could not find the nucleotide sequence data for the genome, despite the fact that the strain under study is registered there under the given accession numbers. If this is an error, it needs to be corrected.
- Page 6, the first paragraph – “One notable species is G. lapidescens, which is distinguished by its distinctive medicinal properties.” – please add a reference.
- Page 6, the first paragraph – “Through rigorous strain isolation and domestication cultivation procedures from these samples, a distinct strain was identified and characterized”. – However, in the section “Materials and methods” there are no methods for isolating the G. lapidescens QL01 strain, and there is no description of the cultivation conditions. Neither the “Materials and methods” section nor the “Results” section provides data on the characteristics of this strain. Text on page 6 -“This strain was designated G. lapidescens QL01. This particular isolate displays unique mycelial morphological characteristics, indicating a substantial deviation from the characteristics of commercially available cultivars. A thorough evaluation has been conducted, which has confirmed the strain's superior and stable medicinal attributes.” – These are general statements that need to be supported either by the data you have obtained (for example, in the form of comparison tables with other fungi) demonstrating the uniqueness of this strain, or by references to published studies.
- Page 8 – figure caption “Figure 2 (A and B) show the SNP and methylation analyses of Gerronema lapidescens QL01 respectively.” – please move it under the drawing.
- Section name “3.3. SNP and Methylation Analysis of Gerronema lapidescens ” It is necessary to write in italics to maintain uniformity.
- There are no axis signatures in Figure 2.
- In Figure 2B, the legend is shifted directly to the field of the drawing; it must be moved beyond the border of the drawing.
- On page 8, the numbering of sections has been knocked down. Section number “ 3.3 Phylogenetic and Gene Family Variation Analysis.” repeats the section number on page 7 “3.3. SNP and Methylation Analysis of Gerronema lapidescens QL01.” Please correct the numbering of sections throughout the manuscript.
- Figure 3 – What is indicated in yellow is impossible to read. Please change the color.
- Page 10 – “Due to their high degree of polymorphism and extensive genomic distribution, they are frequently regarded as ideal molecular markers.” – It is required to add references to studies where this is shown.
- Page 10 - “A comparative SSR analysis of four Auricularia species, including Wolfiporia cocos, was conducted, which revealed that trinucleotide repeats are a major component of SSRs in most Auricularia species, with all exceeding 40%.” – The name of the genus Auricularia should be written in italics. And why are you talking about this genus of fungi? Is it not clear how this information is related to your research? In addition, Wolfiporia cocos does not belong to the genus Auricularia — it is an independent genus of fungi from the family Polyporaceae.
The information provided is questionable.
- The caption to Figure 5 is incorrect. - “Figure 5. This study sets out to identify the core genes involved in the secondary metabolite biosynthesis in Gerronema lapidescens A) An investigation was conducted into the distribution of subgenomic secondary metabolite biosynthesis core genes on chromosomes and sequence fragments. B) Phylogenetic tree analysis of STSs was performed. C) The characteristics of domain of enzymes with multiple domains were examined.” – The figure caption needs to be changed. The caption to the figure should characterize what is shown on the graphs, and not describe the research carried out. Why did you put the purpose of the study as a caption to the figure, rather than writing it at the end of the Introduction section? Is this the main purpose of the article?
- Section name “3.5 Search and analysis of secondary metabolite-related genes” does not match the information contained in this section. This section does not contain any data related to the search and analysis of genes associated with secondary metabolites of the fungus.
- In the Supplemental materials file – please delete the blank pages 16,19.
Author Response
Response to Reviewer: 1
Dear authors,
Your article “The First Whole Genome Sequence and Methylation Profile of Gerronema lapidescens QL01” undoubtedly demonstrates a high degree of novelty, as the manuscript presents, for the first time, the genome sequencing and assembly of the G. lapidescens QL01 strain. However, while reading your article, I had a number of questions and comments.
Dear authors, unfortunately, the version of the manuscript that was provided to me for review lacks line numbering, which somewhat complicates the review process.
Thank you very much for your positive comments on our article, "The First Whole Genome Sequence and Methylation Profile of Gerronema lapidescens QL01."We are delighted that you recognise the novelty of our work. Your questions and comments are highly valued and will be addressed carefully.
We apologise sincerely for the oversight regarding the line numbering in the manuscript version you received. It was an unfortunate omission that we understand made the review process more challenging. Rest assured, we have now included line numbers in the revised version, and we will ensure that this feature is maintained in all subsequent drafts.
Thank you once again for your understanding and support. We look forward to incorporating your feedback to further improve our manuscript.
Q1. The purpose of the research is not stated in the manuscript.
A1. The third paragraph of the Introduction section clearly states the purpose of this study.Please take note of it.
Q2. Page 2 – “Bioactive compounds include G. lapidescens lectin and sterols [7,8], conferring antitumour, antiparasitic, anthelmintic, anti-inflammatory, antiviral, immunomodulatory, and hypoglycaemic activities[9]” – you are referring to a non-existent article published in a non-existent journal. [7] - Yonghai, Y.; Jun, G.; Mingkun, Y. Studies on the purification and properties of Omphalia lapidescens lectin. Systematics of Mycology 2000, 278-282, doi:10.13346/j.mycosystema.2000.02.023. Did you use generative neural networks when writing the article?
A2. Reference 7, Yonghai,Y.; Jun,G.;Mingkun,Y.Studies on the purification and properties of Omphalia lapidescens lectin. Systematics of Mycology 2000,278-282, doi: 10.13346/j.mycosystema. 2000.02.023, is a research paper published in the authoritative Chinese mycology journal Mycosystema (formerly known as Systematics of Mycology). This paper can be found on Europe PMC(https://europepmc.org/article/cba/335741) and Semantic Scholar (Corpus ID: 86993751).
Q3. In the Materials and Methods section, there is no information about the source of the lapidescens QL01 strain used in the research, and there is also no characteristic of the organism.
A3. The first paragraph of the Results section describes the source of the samples. G. lapidescensis a sclerotial fungus, and a photograph of its sclerotia is shown in Figure 1.
Q4. Page 2, Section “2.1 Fungal material and nucleic acid extraction” - “The high-quality mycelia of G. lapidescens QL01 were cultured in PDB liquid medium (25°C, 180 rpm, one week) to obtain mycelium for DNA extraction.” - It is necessary to specify the composition of the PDB nutrient medium or provide a link to the article where it is indicated.
A4. Done.
Q5. Section “2.1 Fungal material and nucleic acid extraction” – Was there any mycelium treatment before using the Fungal Genomic DNA Rapid Extraction Kit (Sangon Biotech, Shanghai, China)?
A5. DNA extraction from mycelium requires pretreatment, such as washing and grinding with liquid nitrogen, but these steps are included in the Fungal Genomic DNA Rapid Extraction Kit.
Q6. In the Materials and Methods section, when mentioning the devices that were used for this or that manipulation, please write the company and the country of the manufacturer.
A6. Revised.
Q7. Page 3 – “…and scaffolds evaluated for continuity, accuracy, consistency, and completeness.” -how was this check conducted? You need to add a reference.
A7. Done.
Q8. Page 3, Section “2.4 SNP site detection” - “Initially, the Bur-rows-Wheeler Transform (BWT) algorithm was employed, specifically the FM-index method” – you need to add references.
A8. Done.
Q9. Page 5, fourth paragraph – “The process of visualization was executed through the utilization of bespoke Python scripts.” – did you use standard scripts or copyrighted ones? If these were author's scripts, then you must provide a link to the repository with the code or the script attachment to the article.
A9. The custom Python scripts are provided in the supplementary material and are described in the manuscript where relevant.
Q10. Section “2.9. Data Availability” – “The final genome assembly and associated datasets for G. lapidescens QL01 have been submitted to NCBI under BioProject PRJNA 1256626, and BioSample SAMN 48188347.” – Using the specified genome assembly data, I was unable to view the genome in the NCBI database. I could not find the nucleotide sequence data for the genome, despite the fact that the strain under study is registered there under the given accession numbers. If this is an error, it needs to be corrected.
A10. The relevant data had been submitted to NCBI prior to submission. However, as of today, NCBI GenBank still shows it as "processing." The NCBI BioProject and BioSample accession numbers are naturally linked to the NCBI GenBank accession number, which can provide clues for it.
Q11. Page 6, the first paragraph – “One notable species is G. lapidescens, which is distinguished by its distinctive medicinal properties.” – please add a reference.
A11. Added.
Q12. Page 6, the first paragraph – “Through rigorous strain isolation and domestication cultivation procedures from these samples, a distinct strain was identified and characterized”. – However, in the section “Materials and methods” there are no methods for isolating the G. lapidescens QL01 strain, and there is no description of the cultivation conditions. Neither the “Materials and methods” section nor the “Results” section provides data on the characteristics of this strain. Text on page 6 -“This strain was designated G. lapidescens QL01. This particular isolate displays unique mycelial morphological characteristics, indicating a substantial deviation from the characteristics of commercially available cultivars. A thorough evaluation has been conducted, which has confirmed the strain's superior and stable medicinal attributes.” – These are general statements that need to be supported either by the data you have obtained (for example, in the form of comparison tables with other fungi) demonstrating the uniqueness of this strain, or by references to published studies.
A12. Added and revised.
Q13. Page 8 – figure caption “Figure 2 (A and B) show the SNP and methylation analyses of Gerronema lapidescens QL01 respectively.” – please move it under the drawing.
A13. Thank you for your valuable suggestion; the manuscript has been accordingly modified.
Q14. Section name “3.3. SNP and Methylation Analysis of Gerronema lapidescens ” It is necessary to write in italics to maintain uniformity.
A14. Thank you for your valuable suggestion; the manuscript has been accordingly modified.
Q15. There are no axis signatures in Figure 2.
A15. The vertical axis represents abundance, with the unit being "1," and the horizontal axis represents chromosome length, with the unit being Mb. These are clearly labeled in the figure.
Q16. In Figure 2B, the legend is shifted directly to the field of the drawing; it must be moved beyond the border of the drawing.
A16. The figure 2 has been amended and inserted into the manuscript.
Q17. On page 8, the numbering of sections has been knocked down. Section number “ 3.3 Phylogenetic and Gene Family Variation Analysis.” repeats the section number on page 7 “3.3. SNP and Methylation Analysis of Gerronema lapidescens QL01.” Please correct the numbering of sections throughout the manuscript.
A17. Thank you for your valuable suggestion; the manuscript has been accordingly modified.
Q18. Figure 3 – What is indicated in yellow is impossible to read. Please change the color.
A18. The figure 3 has been amended and inserted into the manuscript.
Q19. Page 10 – “Due to their high degree of polymorphism and extensive genomic distribution, they are frequently regarded as ideal molecular markers.” – It is required to add references to studies where this is shown.
A19. Added.
Q20. Page 10 - “A comparative SSR analysis of four Auricularia species, including Wolfiporia cocos, was conducted, which revealed that trinucleotide repeats are a major component of SSRs in most Auricularia species, with all exceeding 40%.” – The name of the genus Auricularia should be written in italics. And why are you talking about this genus of fungi? Is it not clear how this information is related to your research? In addition, Wolfiporia cocos does not belong to the genus Auricularia — it is an independent genus of fungi from the family Polyporaceae. The information provided is questionable.
A20. Thank you for your careful reading. Any inappropriate parts have been corrected.
Q21. The caption to Figure 5 is incorrect. - “Figure 5. This study sets out to identify the core genes involved in the secondary metabolite biosynthesis in Gerronema lapidescens A) An investigation was conducted into the distribution of subgenomic secondary metabolite biosynthesis core genes on chromosomes and sequence fragments. B) Phylogenetic tree analysis of STSs was performed. C) The characteristics of domain of enzymes with multiple domains were examined.” – The figure caption needs to be changed. The caption to the figure should characterize what is shown on the graphs, and not describe the research carried out. Why did you put the purpose of the study as a caption to the figure, rather than writing it at the end of the Introduction section? Is this the main purpose of the article?
A21. Revised and thank you.
Q22. Section name “3.5 Search and analysis of secondary metabolite-related genes” does not match the information contained in this section. This section does not contain any data related to the search and analysis of genes associated with secondary metabolites of the fungus.
A22. Revised.
Q23. In the Supplemental materials file – please delete the blank pages 16,19.
A23. Thank you for your valuable suggestion; the manuscript has been accordingly modified.
Reviewer 2 Report
This manuscript presents the first whole-genome sequence and methylation profile of Gerronema lapidescens QL01, along with a comprehensive analysis of its biosynthetic potential for secondary metabolites, environmental adaptability, genetic diversity, and CAZyme composition.
As G. lapidescens is a rare and previously uncharacterized fungal resource with reported pharmacological potential, this study has the potential to serve as a valuable reference for researchers in related fields.
However, in its current form, the manuscript presents certain mismatches between the stated objectives, the methodological approach, and the way the findings are discussed. To improve its scientific rigor and clarity, substantial revisions are necessary.
In particular, the relationships between DNA methylation and environmental adaptation, the purpose and sample details of the SNP analysis, the rationale behind reference genome selection, and issues related to taxonomic nomenclature should be addressed more explicitly and systematically.
If the authors are able to address these points and better align the study’s components, the manuscript could become a worthwhile contribution to the area of research.
Please refer to the detailed comments below.
- One of the key goals of the study appears to be understanding how DNA methylation could contribute to the environmental or evolutionary adaptation of wild fungal species. However, the methylation analysis in the results is primarily limited to its potential role in regulating silent biosynthetic gene clusters (BGCs). There is a noticeable lack of discussion on how lapidescens may utilize epigenetic regulation for survival in extreme or lithic environments. This represents a disconnect between the initial research aim and the actual interpretation of results. A clearer explanation or expanded analysis addressing environmental adaptation mechanisms is recommended to improve scientific coherence.
- In Section 3.1, the authors mention that Gerronema lapidescens QL01 was isolated during the collection of Bolbitius However, the phylogenetic relationship between these two genera is not clearly explained. While Bolbitius and Gerronema are both classified within the order Agaricales, they are generally thought to belong to separate phylogenetic lineages. Moreover, Bolbitius species are not included in the phylogenetic tree presented in the study, making the rationale behind this statement unclear. The authors should clarify how the isolate was identified as G. lapidescens—including morphological characteristics and molecular evidence (e.g., ITS, LSU)—and confirm that there is no taxonomic confusion with Bolbitius. If such clarification cannot be provided, the use of the term “Bolbitius resource” may be misleading and should either be revised or removed.
- The authors report the use of Ganoderma lucidum QL01 and other Ganoderma species as reference genomes for homology-based gene prediction using GeMoMa v1.7. However, the target organism, Gerronema lapidescens QL01, is phylogenetically distinct from Ganoderma Indeed, the phylogenetic analysis presented in Section 3.4 emphasizes a closer relationship with Lentinula edodes and Dendrothele bispora, rather than with Ganoderma. As Ganoderma is typically a wood-decaying fungus with potentially divergent genomic architecture and ecological niche, the rationale behind selecting G. lucidum QL01 as a reference is unclear. The authors should clarify the scientific reasoning for this choice and explain why it was considered appropriate over other, more closely related taxa. Alternatively, this may be a typographical error that needs correction.
- In Section 3.3, the authors report the identification of 8,460,308 SNPs within the genome of lapidescens QL01. However, the manuscript does not make it fully clear what the authors hoped to uncover through the SNP analysis or how these findings should be interpreted biologically. It is not stated whether these SNPs are located within functional gene regions associated with particular phenotypes (e.g., environmental stress tolerance or secondary metabolite production), or whether the analysis was simply exploratory in nature, intended to quantify general genomic variation. Furthermore, no comparative analysis (e.g., with closely related species or control strains) is provided to contextualize the SNP distribution, and the interpretation appears limited to basic descriptive statistics. To make this section more meaningful and easier to follow, the authors should more clearly explain the reasoning behind the SNP analysis and its biological implications.
- The details regarding sample size and biological replication in the SNP analysis lack clarity. It remains ambiguous whether the reported SNPs were derived exclusively from a single strain (G. lapidescens QL01) using Illumina short-read sequencing, or if the analysis encompassed multiple isolates to evaluate population-level genetic variation. In the latter scenario, the authors should comprehensively provide critical metadata, including the precise sample size (n), specific collection sites, and detailed experimental conditions. Conversely, if the analysis is based on a single genome, the authors must explicitly state that the identified SNPs represent intra-genomic heterozygosity and critically discuss the inherent limitations of interpreting such data without comparative or replicate context. Addressing this methodological transparency is crucial for accurately assessing the biological significance of the findings and determining the extent to which the results can be generalized.
Author Response
Response to Reviewer: 2
This manuscript presents the first whole-genome sequence and methylation profile of Gerronema lapidescens QL01, along with a comprehensive analysis of its biosynthetic potential for secondary metabolites, environmental adaptability, genetic diversity, and CAZyme composition.
As G. lapidescens is a rare and previously uncharacterized fungal resource with reported pharmacological potential, this study has the potential to serve as a valuable reference for researchers in related fields.
However, in its current form, the manuscript presents certain mismatches between the stated objectives, the methodological approach, and the way the findings are discussed. To improve its scientific rigor and clarity, substantial revisions are necessary.
In particular, the relationships between DNA methylation and environmental adaptation, the purpose and sample details of the SNP analysis, the rationale behind reference genome selection, and issues related to taxonomic nomenclature should be addressed more explicitly and systematically.
If the authors are able to address these points and better align the study’s components, the manuscript could become a worthwhile contribution to the area of research.
Please refer to the detailed comments below.
Thank you very much for your thorough review and constructive comments on our manuscript. We are grateful for your recognition of the potential significance of our study on the whole-genome sequence and methylation profile of Gerronema lapidescens QL01.Your suggestions are highly valuable and will significantly enhance the scientific rigor and clarity of our manuscript.
We are determined to align the components of our study more effectively and present a cohesive and scientifically robust manuscript. We will carefully review each of your detailed comments and implement the necessary changes to address them fully.
Thank you once again for your time and valuable feedback. We look forward to submitting a revised version of our manuscript that meets the high standards of your journal.
Q1. One of the key goals of the study appears to be understanding how DNA methylation could contribute to the environmental or evolutionary adaptation of wild fungal species. However, the methylation analysis in the results is primarily limited to its potential role in regulating silent biosynthetic gene clusters (BGCs). There is a noticeable lack of discussion on how lapidescens may utilize epigenetic regulation for survival in extreme or lithic environments. This represents a disconnect between the initial research aim and the actual interpretation of results. A clearer explanation or expanded analysis addressing environmental adaptation mechanisms is recommended to improve scientific coherence.
A1. Thank you very much for your thoughtful consideration and for highlighting the importance of methylation in our research. Indeed, our primary goal for this manuscript was to provide the first genomic information for Gerronema lapidescens. Building on this, we characterized the methylation profile of its genome (Results, Section 3). Given that Gerronema lapidescens is a traditional Chinese medicine, and the abundance of biosynthetic gene clusters (BGCs) in its genome does not match its pharmacological properties, we therefore performed a methylation analysis of its BGCs (Results, Section 7). We understand your expectations, and this is equally important to us. The revised article will fully address your concerns, and future research will focus on exploring the relationship between methylation and environmental adaptation.
Q2. In Section 3.1, the authors mention that Gerronema lapidescens QL01 was isolated during the collection of Bolbitius However, the phylogenetic relationship between these two genera is not clearly explained. While Bolbitius and Gerronema are both classified within the order Agaricales, they are generally thought to belong to separate phylogenetic lineages. Moreover, Bolbitius species are not included in the phylogenetic tree presented in the study, making the rationale behind this statement unclear. The authors should clarify how the isolate was identified as G. lapidescens—including morphological characteristics and molecular evidence (e.g., ITS, LSU)—and confirm that there is no taxonomic confusion with Bolbitius. If such clarification cannot be provided, the use of the term “Bolbitius resource” may be misleading and should either be revised or removed.
A2. This is a misunderstanding caused by an oversight. The title should have been "Collection and Domestication Cultivation of Wild Macrofungi Resources in the Qinling Mountains", but has now been corrected.
Q3. The authors report the use of Ganoderma lucidum QL01 and other Ganoderma species as reference genomes for homology-based gene prediction using GeMoMa v1.7. However, the target organism, Gerronema lapidescens QL01, is phylogenetically distinct from Ganoderma Indeed, the phylogenetic analysis presented in Section 3.4 emphasizes a closer relationship with Lentinula edodes and Dendrothele bispora, rather than with Ganoderma. As Ganoderma is typically a wood-decaying fungus with potentially divergent genomic architecture and ecological niche, the rationale behind selecting G. lucidum QL01 as a reference is unclear. The authors should clarify the scientific reasoning for this choice and explain why it was considered appropriate over other, more closely related taxa. Alternatively, this may be a typographical error that needs correction.
A3. As you pointed out, that was a typographical error. Thank you for your careful reading.
Q4. In Section 3.3, the authors report the identification of 8,460,308 SNPs within the genome of lapidescens QL01. However, the manuscript does not make it fully clear what the authors hoped to uncover through the SNP analysis or how these findings should be interpreted biologically. It is not stated whether these SNPs are located within functional gene regions associated with particular phenotypes (e.g., environmental stress tolerance or secondary metabolite production), or whether the analysis was simply exploratory in nature, intended to quantify general genomic variation. Furthermore, no comparative analysis (e.g., with closely related species or control strains) is provided to contextualize the SNP distribution, and the interpretation appears limited to basic descriptive statistics. To make this section more meaningful and easier to follow, the authors should more clearly explain the reasoning behind the SNP analysis and its biological implications.
A4. Thank you for highlighting this shortcoming in our manuscript. You are correct that the general genomic variation quantified through the SNP analysis was not thoroughly discussed. We have addressed this deficiency in the revised version. We agree that comparing SNP distribution with closely related or control strains is a valuable approach. However, at the genus level, there are no available genomes for species closely related to G. lapidescens. While there are some genomes from related species at the family level, we were unable to obtain the raw sequencing data necessary for a comprehensive comparative analysis.
Q5. The details regarding sample size and biological replication in the SNP analysis lack clarity. It remains ambiguous whether the reported SNPs were derived exclusively from a single strain (G. lapidescens QL01) using Illumina short-read sequencing, or if the analysis encompassed multiple isolates to evaluate population-level genetic variation. In the latter scenario, the authors should comprehensively provide critical metadata, including the precise sample size (n), specific collection sites, and detailed experimental conditions. Conversely, if the analysis is based on a single genome, the authors must explicitly state that the identified SNPs represent intra-genomic heterozygosity and critically discuss the inherent limitations of interpreting such data without comparative or replicate context. Addressing this methodological transparency is crucial for accurately assessing the biological significance of the findings and determining the extent to which the results can be generalized.
A5. The SNP analysis detailed in our manuscript was conducted on the sequenced strain of G. lapidescens QL01 mentioned in the manuscript. The raw data utilised were from the Illumina short-read sequencing employed to assist in the genome assembly. The lack of methodological transparency has been rectified in the revised version. As for the significance of the SNP analysis, which is similar to the previous question, we have thoroughly discussed it in the discussion section.
Round 2
Reviewer 1 Report
Dear authors,
Thank you for making a number of edits to your manuscript, but I still have a few comment.
- Supplemental Material – there are no units of measurement on the axes of the graphs.
- In your response to the review of the first version of the article regarding the presence of a purpose, you mentioned that “The third paragraph of the Introduction section clearly states the purpose of this study,” however, I did not find the purpose of the article in the third paragraph of the Introduction. You write — Lines 69-72 — “Specifically, we aimed to elucidate its genetic architecture and metabolite composition, investigate the genes involved in medicinal compound biosynthesis and environmental adaptation, and pioneer the in-silico prediction of core gene methylation. These foundations will catalyse new research avenues and applications for G. lapidescens.” — these are the objectives, not one overall purpose of the study. Please formulate a clear research purpose.
(As an option: The purpose of this study is to investigate the genetic architecture, metabolite composition, and adaptation mechanisms of G. lapidescens to develop new directions and expand the practical use of G. lapidescens.) - Line 139 – “2.4 SNP site detection” – the abbreviation should be spelled out at its first mention in the text.
- Line 213 – (Smith et al., 2022). Please correct the reference formatting according to the journal’s guidelines.
- Figure 2 – there are no axis labels, so it is unclear what is measured on the vertical (Y) and horizontal (X) axes. Labels and units must be added. The color scale to the right of graph B is too small and not explained in words.
Suggested axis formatting:
For graph A (SNP):
X axis: "Genome position (Mb)" or "Position on chromosome (Mb)".
Y axis: "Number of SNPs" or "SNP density" (if density, specify "SNPs per window", e.g., "SNPs per 10 kb window").
For graph B (methylation):
X axis: "Genome position (Mb)" or "Position on chromosome (Mb)".
Y axis: "Number of methylated sites" or "Methylation level" (depending on what is shown).
Add a caption to the color scale, e.g., "Number of methylated sites" or "Methylation count". - Figure 4 – there are no axis labels on Figure 4B.
- Figure 5 – there are no axis labels on Figure 5C.
- Table 1 – please change the table title. The table title should reflect the nature and content of the presented data, not the results obtained.
- Table 1 – in some rows, multiple genes are listed in one cell separated by spaces (e.g., cluster 3: geneGsp00577 Terpene geneGsp00579 Ripp-like), although according to the headers, one gene should correspond to one place.
I recommend formatting the table by using tabs or another clear delimiter. If there are multiple genes in one cluster, either use multiple rows for each gene or add an additional column for the second gene. Otherwise, such shifts in the table confuse the reader. - Figure 5A – according to Table 1 data, chromosome Chr1 should contain 9 genes with Core Gene IDs: geneGsp00150, geneGsp00338, geneGsp00577, geneGsp00579, geneGsp00614, geneGsp00678, geneGsp00859, geneGsp00862, geneGsp00921, however geneGsp00921 is missing on Figure 5A.
Figure 5A – chromosome Chr3 is missing geneGsp02067 and geneGsp02153, which are present in Table 1.
Figure 5A – chromosome Chr7 is missing geneGsp07102, which is present in Table 1.
Figure 5A – chromosome Chr8 is missing geneGsp08613, which is present in Table 1. - Line 472 – “L. and bispora.” Please replace with “L. bispora и D. bispora.”
- Line 496 – “The” – remove italics.
Author Response
Reviewer 1
Are all figures and tables clear and well-presented?
No
Some figures do not have axis captions, the name of Table 1 needs to be changed, Table 1 needs to be formatted and structured for clearer understanding. There are inconsistencies in Figure 5A with the data in Table 1.
Response: We are indebted to you for your meticulous eye and the time you have invested. In direct response to your observations, we have now supplied complete axis captions for every figure, retitled Table 1 to reflect its contents accurately, and reformatted the table for immediate clarity. The inconsistencies between Figure 5A and Table 1 have been reconciled by re-plotting the figure with the verified data points. We believe these revisions render both figures and tables fully clear and consistent.
Detailed comments
Comments1: Supplemental Material – there are no units of measurement on the axes of the graphs.
Response 1:The supplementary material includes three additional figures, all of which have clear labels on the x-axis and y-axis. These figures are generated by software, with corresponding units of 1 or 1%, which are generally not labeled. This is the case for almost all genome-related articles, and there is no strict requirement for it.
Comments 2: In your response to the review of the first version of the article regarding the presence of a purpose, you mentioned that “The third paragraph of the Introduction section clearly states the purpose of this study,” however, I did not find the purpose of the article in the third paragraph of the Introduction. You write — Lines 69-72 — “Specifically, we aimed to elucidate its genetic architecture and metabolite composition, investigate the genes involved in medicinal compound biosynthesis and environmental adaptation, and pioneer the in-silico prediction of core gene methylation. These foundations will catalyse new research avenues and applications for G. lapidescens.” — these are the objectives, not one overall purpose of the study. Please formulate a clear research purpose. (As an option: The purpose of this study is to investigate the genetic architecture, metabolite composition, and adaptation mechanisms of G. lapidescens to develop new directions and expand the practical use of G. lapidescens.)
Response 2:Adopted and corrected
Comments 3: Line 139 – “2.4 SNP site detection” – the abbreviation should be spelled out at its first mention in the text.
Response 3:The SNP, which first appeared in the abstract, has been supplemented as required.
Comments 4: Line 213 – (Smith et al., 2022). Please correct the reference formatting according to the journal’s guidelines.
Response 4:Modified.
Comments 5: Figure 2 – there are no axis labels, so it is unclear what is measured on the vertical (Y) and horizontal (X) axes. Labels and units must be added. The color scale to the right of graph B is too small and not explained in words.
Suggested axis formatting:
For graph A (SNP):
X axis: "Genome position (Mb)" or "Position on chromosome (Mb)".
Y axis: "Number of SNPs" or "SNP density" (if density, specify "SNPs per window", e.g., "SNPs per 10 kb window").
For graph B (methylation):
X axis: "Genome position (Mb)" or "Position on chromosome (Mb)".
Y axis: "Number of methylated sites" or "Methylation level" (depending on what is shown).
Add a caption to the color scale, e.g., "Number of methylated sites" or "Methylation count".
Response 4:Revised and thank you.
Comments 5: Figure 4 – there are no axis labels on Figure 4B.
Response 5:Added.
Comments 6: Figure 5 – there are no axis labels on Figure 5C.
Response 6:Added.
Comments 7: Table 1 – please change the table title. The table title should reflect the nature and content of the presented data, not the results obtained.
Response 7:Revised and thank you.
Comments 8: Table 1 – in some rows, multiple genes are listed in one cell separated by spaces (e.g., cluster 3: geneGsp00577 Terpene geneGsp00579 Ripp-like), although according to the headers, one gene should correspond to one place.
Response 8:The first column represents the cluster number, and the fourth column indicates the core gene ID. It is normal for a gene cluster to have two core genes, and this presentation is standard.
Comments 9: I recommend formatting the table by using tabs or another clear delimiter. If there are multiple genes in one cluster, either use multiple rows for each gene or add an additional column for the second gene. Otherwise, such shifts in the table confuse the reader.
Response 9:Thank you for your meticulous consideration. We have made minor adjustments to the presentation of Table 1 according to your suggestions to make it more comprehensible for readers. Additionally, we have added notes in the table to aid understanding.
Comments 10: Figure 5A – according to Table 1 data, chromosome Chr1 should contain 9 genes with Core Gene IDs: geneGsp00150, geneGsp00338, geneGsp00577, geneGsp00579, geneGsp00614, geneGsp00678, geneGsp00859, geneGsp00862, geneGsp00921, however geneGsp00921 is missing on Figure 5A.
Figure 5A – chromosome Chr3 is missing geneGsp02067 and geneGsp02153, which are present in Table 1.
Figure 5A – chromosome Chr7 is missing geneGsp07102, which is present in Table 1.
Figure 5A – chromosome Chr8 is missing geneGsp08613, which is present in Table 1.
Response 10:Thank you for your careful checking and verification. The missing information and inappropriate aspects in the figures have been revised and corrected.
Comments 11: Line 472 – “L. and bispora.” Please replace with “L. bispora и D. bispora.”
Response 11:Revised.
Comments 12: Line 496 – “The” – remove italics.
Response 12:Done.
Reviewer 2 Report
The authors have generally addressed the major concerns raised in the first review, and the manuscript has improved in clarity and overall quality. The revisions to the methylation analysis, phylogenetic clarification, and reference genome selection are satisfactory.
However, the interpretation of the SNP analysis remains unclear. It is not explicitly stated whether the reported SNPs represent intra-genomic heterozygosity within the single sequenced strain (QL01) or whether they were derived from NGS analysis of multiple isolates at the population level. To avoid misinterpretation, the authors should clearly define the scope of the SNP analysis. If the data are based on a single genome, the limitations of such an approach (i.e., not representing population-level variation) must be explicitly acknowledged.
In summary, while the manuscript has been substantially improved and most concerns have been addressed, greater methodological transparency and clarification regarding the SNP analysis are still required.
Please refer to the Major Comment.
Author Response
Reviewer 2
The authors have generally addressed the major concerns raised in the first review, and the manuscript has improved in clarity and overall quality. The revisions to the methylation analysis, phylogenetic clarification, and reference genome selection are satisfactory.
However, the interpretation of the SNP analysis remains unclear. It is not explicitly stated whether the reported SNPs represent intra-genomic heterozygosity within the single sequenced strain (QL01) or whether they were derived from NGS analysis of multiple isolates at the population level. To avoid misinterpretation, the authors should clearly define the scope of the SNP analysis. If the data are based on a single genome, the limitations of such an approach (i.e., not representing population-level variation) must be explicitly acknowledged.
In summary, while the manuscript has been substantially improved and most concerns have been addressed, greater methodological transparency and clarification regarding the SNP analysis are still required.
Response :Thank you for recognising the improvements we have made. In direct response to your final point, we have inserted an explicit paragraph in the Discussion (second paragraph) that clarifies the scope of the SNP analysis: all variants were called from the single sequenced strain QL01 and therefore reflect intra-genomic heterozygosity rather than population-level diversity.
Round 3
Reviewer 1 Report
Dear authors!
Thank you for your careful and high-quality revision of the manuscript. Thanks to your meticulous approach and the work done, the article has gained clarity, structure and a high level of presentation.
The manuscript meets the quality criteria and is ready for publication.
Reviewer 2 Report
The authors have carefully addressed the points raised in my previous review. In particular, they have clarified the scope of the SNP analysis by explicitly stating that all variants were derived from the single sequenced strain QL01, thereby reflecting intra-genomic heterozygosity rather than population-level diversity. This important clarification has been incorporated into the revised Discussion section.
Overall, the manuscript has been sufficiently improved and the revisions adequately address the concerns raised. I consider the current version suitable for publication in the Journal of Fungi.
Please refer to the major comment.